# Mechanical Sensing Element PDLIM5 Promotes Osteogenesis of Human Fibroblasts by Affecting the Activity of Microfilaments

**DOI:** 10.3390/biom11050759

**Published:** 2021-05-19

**Authors:** Xiaolan Huang, Rongmei Qu, Yan Peng, Yuchao Yang, Tingyu Fan, Bing Sun, Asmat Ullah Khan, Shutong Wu, Kuanhai Wei, Chujiang Xu, Jingxing Dai, Jun Ouyang, Shizhen Zhong

**Affiliations:** 1Guangdong Provincial Key Laboratory of Medical Biomechanics & Department of Anatomy, School of Basic Medical Science, Southern Medical University, Guangzhou 510515, China; huangxiaolan2021@gmail.com (X.H.); shirth@smu.edu.cn (R.Q.); pengyan9306@gmail.com (Y.P.); gzyyc27@gmail.com (Y.Y.); fty1886@gmail.com (T.F.); bingsun06@gmail.com (B.S.); khan9905@smu.edu.cn (A.U.K.); wushutong1123@gmail.com (S.W.); 2Division of Orthopaedics and Traumatology, Department of Orthopaedics, Guangdong Provincial Key Laboratory of Bone and Cartilage Regeneration Medicine, Nanfang Hospital, Southern Medical University, Guangzhou 510515, China; gzkhwei@163.com; 3Department of Orthopedics, TCM-Integrated Hospital, Southern Medical University, Guangzhou 510000, China; chujiang7955@163.com

**Keywords:** human skin fibroblasts, osteogenesis, PDLIM5, microfilament, mechanical sensing element

## Abstract

Human skin fibroblasts (HSFs) approximate the multidirectional differentiation potential of mesenchymal stem cells, so they are often used in differentiation, cell cultures, and injury repair. They are an important seed source in the field of bone tissue engineering. However, there are a few studies describing the mechanism of osteogenic differentiation of HSFs. Here, osteogenic induction medium was used to induce fibroblasts to differentiate into osteoblasts, and the role of the mechanical sensitive element PDLIM5 in microfilament-mediated osteogenic differentiation of human fibroblasts was evaluated. The depolymerization of microfilaments inhibited the expression of osteogenesis-related proteins and alkaline phosphatase activity of HSFs, while the polymerization of microfilaments enhanced the osteogenic differentiation of HSFs. The evaluation of potential protein molecules affecting changes in microfilaments showed that during the osteogenic differentiation of HSFs, the expression of PDLIM5 increased with increasing induction time, and decreased under the state of microfilament depolymerization. Lentivirus-mediated *PDLIM5* knockdown by shRNA weakened the osteogenic differentiation ability of HSFs and inhibited the expression and morphological changes of microfilament protein. The inhibitory effect of knocking down PDLIM5 on HSF osteogenic differentiation was reversed by a microfilament stabilizer. Taken together, these data suggest that PDLIM5 can mediate the osteogenic differentiation of fibroblasts by affecting the formation and polymerization of microfilaments.

## 1. Introduction

Regenerative medicine is a field that studies the repair and regeneration of damaged tissues and organs [1,2]. At present, the repair process of bone trauma diseases involves the use of stem cells [3,4,5], and osteoblasts play an important role in the healing process. Among the seed cells from many sources, studies have shown that human fibroblasts exhibit stem cell characteristics and can differentiate into chondrocytes, osteoblasts, and adipocytes [6,7]. Fibroblasts can further differentiate into osteoblasts and maintain a stable phenotype, and play a similar role in osteogenic differentiation as stem cells [8,9], which are widely used as an ideal source of cell seeds in bone tissue engineering [10,11]. However, although human fibroblasts have potential therapeutic value, the basic mechanism of osteogenic differentiation remains to be fully elucidated. Therefore, the study of the molecular mechanism involved in osteogenic differentiation of fibroblasts can contribute to a better understanding of the stem cell potential of multi-directional differentiation of fibroblasts and lays a solid foundation for its clinical application.

PDLIM5 (also known as ENH) was first discovered in 1996 by Kuroda et al., using yeast two-hybrid technology with protein kinase C (PKC) as the bait protein [12]. PDLIM5, a PDZ-LIM family member, functions as a cytoskeleton adaptor protein. Recent studies have reported that PDLIM5 may be involved in the progression of multiple tumor types [13,14,15,16], and PDLIM5 may also expand its function by remodeling the cytoskeleton [17]. Advances in the life sciences have led to a deeper understanding of the structure of PDLIM5 protein and its functional roles. PDZ-LIM protein plays a role in the development and maintenance of skeletal muscle by stabilizing actin cytoskeleton and regulating cell adhesion through interaction with α-actinin and integrin (αPS2,βPS, β1-integrin) [18,19,20] and as a muscle-specific scaffold protein to tether PKCβ molecules to the Z-disk region of sarcomere [21]. PDLIM5 is involved in the regulation of proliferation and differentiation of myoblasts as a target signal of MicroRNA-17-92 [22]. In addition, it has been reported that PDLIM5 promotes myogenic gene expression in C2C12 cells, thus promoting their myogenic differentiation [23]. Further, PDLIM5 can promote the migration, proliferation, and invasion of the thyroid papillary carcinoma cell line PTC through the RAS-ERK signaling pathway [24]. As a mechanical sensitive element, PDLIM5 shows obvious subcellular localization in mesenchymal stem cells and exhibits tension-dependent shifting in cells [25]. Furthermore, it regulates the mechanical conduction of the YAP pathway [26]. As a new substrate of AMPK at the ser-177 residue, PDLIM5 promotes its phosphorylation and mediates cell migration by inhibiting the Rac1-Arp2/3 pathway [27]. Therefore, as a cytoskeleton-related protein, PDLIM5 plays an important role in the genesis and development of cells.

The cytoskeleton is an important cellular component and plays an important role in the maintenance of cell movement, the reception of external signals, and the effective maintenance of cell internal structure. There are three basic structures of cytoskeleton: microfilaments, intermediate filaments, and microtubules [28,29,30]. Actin is considered to be a key factor in mediating intracellular and extracellular signal responses [31,32]. Studies have reported that microfilaments play an important regulatory role in the process of osteoblastic differentiation, promoting the osteoblastic differentiation of MC3T3 cells through the p38-MAPK signaling pathway [33,34]. By modulating BMP2-Smad signaling and the expression of this signaling downstream target gene *RUNX2* by regulating the aggregation state of microfilaments, thus affect osteoblastic differentiation of osteoblasts [35,36]. It is known that PDLIM5 plays an important role in cardiovascular system [37], nervous system [38], and tumor system [39]. However, its role in fibroblasts has not been specifically elucidated. Therefore, understanding the molecular mechanism of PDLIM5 in osteoblastic differentiation in fibroblasts will help to understand the potential of osteogenic differentiation in fibroblasts and provide researchers with seed-derived cells for studying bone-filling materials.

In this study, human fibroblasts were used as the research model, in which the stability of microfilaments was disturbed by the exposure to treatment with Jasplakinolide (JAS, actin polymerization stabilizer) and Cytochalasin D (CytoD, actin polymerization inhibitor). ALP staining, Western blotting, quantitative real-time polymerase chain reaction (qRT-PCR), and immunofluorescence staining were used to detect the effects of microfilaments on osteoblast differentiation of human fibroblasts. Lentivirus transfection was adopted to down-regulate the expression of the cytoskeleton-related protein PDLIM5 to detect its regulatory effect on microfilaments. Finally, we attempt to explain the molecular mechanism underlying PDLIM5 activity in regulating the osteogenic differentiation of fibroblasts by mediating microfilaments.

## 2. Materials and Methods

### 2.1. Cell Culture, Osteogenic Differentiation, and Treatments

Human skin fibroblasts (HSFs) were obtained from skin dermis, which were purchased from the cell bank of Zhongqiaoxinzhou Company (shanghai; China), and human adipose stem cells (hASCs) were extracted from subcutaneous adipose tissue of healthy individuals, using the type I collagenase digestion method as described previously [40]: adipose tissue was digested at 37 °C with 0.1% (*w*/*v*) type I collagenase to three times the volume of adipose tissue. After 40–60 min, digestion was terminated with Dulbecco’s modified Eagle medium (DMEM) to which 10% fetal bovine serum, 100 U/mL penicillin, and 100 µg/mL streptomycin were added. After centrifuged at 1000 rpm for 10 min, isolated cells were suspended in growth medium (GM) and incubated at 37 °C in 5% CO_2_. Human adipose tissue was donated from the Plastic Surgery Department of Nanfang Hospital. Cells were cultured in growth medium consisting of high-glucose Dulbecco’s modified Eagle’s medium (DMEM; Gibco, Carlsbad, CA, USA) with 10% fetal bovine serum (FBS, Gibco, Carlsbad, CA, USA) and 1% penicillin/streptomycin (Gibco). The growth medium was changed every 2 days.

The cells with 7000/cm^2^ density were seeded on a Petri dish. When the cell confluence reached 80%, the growth medium was replaced by the osteogenic differentiation medium (OS), which contained 10% FBS, 1% penicillin/streptomycin, 100 nM, dexamethasone, 37.5 mg/L of ascorbic acid, 10 mM β-glycerophosphate sodium, and 10 nM VitD3. The medium was changed every 2 days. On days 1, 4, 7, 14, and 21 of culture samples were taken for Western blotting, qRT-PCR, and immunofluorescence analyses to detect the osteogenic differentiation ability and related indexes.

CytoD (0.1 µg/mL) and JAS (10 nM/mL) were added to the osteogenic differentiation medium, and the corresponding growth medium was replaced every day. The samples were treated on day 7 of culture.

### 2.2. Western Blotting

A protein extraction kit (Whole protein extraction kit; Key GEN BioTECH, Nanjing, China) was used to prepare the cell lysis buffer according to the manufacturer’s instructions. Adherent cells were washed with precooled PBS three times, and the lysate was collected following cell lysis for 30 min on ice. The samples were centrifuged at 12,000× *g* at 4 °C for 10 min, which was followed by boiling for 5 min at 100 °C. Similarly, nuclear protein extraction was performed using the cytoplasmic-nuclear protein extraction kit (nucleoprotein and cytoplasmic protein extraction kit, Key GEN BioTECH, Nanjing, China) to prepare cytoplasmic and nuclear lysis buffers. The cell precipitates were digested by trypsin and the cytoplasmic lysis buffer was added for 30 min on ice. The supernatant containing the cytoplasmic protein was collected after centrifugation. The nuclear lysis buffer was added to the remaining precipitate and the lysate was collected after precipitates were lysed for 30–60 min at 4 °C. The supernatant containing the nuclear protein was collected after centrifugation at 12,000× *g* at 4 °C for 30 min, followed by boiling at 100 °C for 5 min.

The 10% SDS-PAGE protocol was used to separate the proteins, which were then transferred to a PVDF membrane (Millipore, Waltham, MA, USA). The PVDF membrane was blocked for 1 h with 5% milk in tris-buffer salt solution (TBS) containing 0.1% Tween-20. The membranes were then incubated with the following primary antibodies: anti-PDLIM5 antibody (dilution 1:2000; ab85967), anti-α-actinin1 antibody (dilution 1:1500; ab68194), anti-osteopontin antibody (dilution 1:1500; ab69498), and anti-alkaline phosphatase (ALP) antibody (dilution 1:1000; ab126820), which were purchased from Abcam (Cambridge, UK). The PDLIM5 monoclonal antibody (dilution 1:1000; H00010611-M01) was purchased from Abnova Company (Taipei, Taiwan). The anti-glyceraldehyde 3-phosphate dehydrogenase (GAPDH) antibody (dilution 1:5000; AP0063) was purchased from Bioworld Company (Bloomington, MN, USA). The anti-RUNT-related transcription factor 2 (RUNX2) antibody (dilution 1:1000; 12556S), anti-β-actin antibody (dilution 1:2000; 4970S), and the anti-YAP antibody (dilution 1:1000; 14074S) were purchased from Cell Signaling Technology (Danvers, MA, USA). The membranes were washed for 10 min in TBST three times and then incubated for 1 h with horseradish peroxidase (HRP)-conjugated secondary antibodies (dilution 1:5000; Fudebio, Hangzhou, China) in 5% milk at room temperature. After washes with TBST, the membrane was visualized using FDbio-Dura ECL kit (Fudebio, Hangzhou, China). The immunoreactive bands were quantitatively analyzed using ImageJ software (V1.4.3.64; National Institutes of Health; Bethesda, MD, USA), and the relative expression of each immunoreactive band was normalized to GAPDH.

### 2.3. RNA Extraction and Quantitative Real-Time PCR 

TRIzol reagent (Invitrogen, Carlsbad, CA, USA) was used to extract the total RNA according to the manufacturer’s instructions. Total RNA (1 μg) was reverse transcribed to cDNA, using RevertAid First Strand cDNA synthesis kit, (Thermo Fisher, Waltham, MA, USA). Quantitative real-time polymerase chain reaction (qPCR) was performed using ABI StepOne Plus System (American Applied Biology Systems Inc. Waltham, MA, USA) with a fluorescent labeled SYBR dye in triplicate, using specific primers for *RUNX2, ALP, OPN, β-actin, α-actinin1, PDLIM5, and GAPDH*, which was used as an endogenous control. All primers used for the analysis are listed in Table 1. Calculation of relative expression of different gene transcripts was performed using the 2^−ΔΔCt^ method.

### 2.4. Immunofluorescence

The cells inoculated on the climbing piece were washed three times with serum-free DMEM basic medium, then fixed in 4% paraformaldehyde at 25 °C for 10 min and washed three times with PBS. Cells were then permeated for 5 min at 25 °C in 0.1% TritonX-100 in PBS and then blocked with phosphate buffer (PBS) containing 2% bovine serum albumin (BSA) for 1 h at 25 °C. The samples were incubated overnight with the following primary antibodies at 4 °C: PDLIM5 monoclonal antibody (dilution 1:500; H00010611-M01), anti-α-actinin1 antibody (dilution, 1:500; ab68194), and anti-YAP antibody (dilution 1:500; 14074S). After washing three times with PBST, the cells were stained with the corresponding secondary antibody avoiding light sources. The secondary antibodies used were AlexaFluor488 goat anti-mouse (1:500), Cy3 labeled goat anti-rabbit (1:500), Cy3 labeled goat anti-mouse (1:500), all purchased from Beyotime Company (shanghai, China). Alexa Fluor 568 phalloidin (1:500) was purchased from Invitrogen (Carlsbad, CA, USA). Finally, 4′,6-diamidino-2-phenylindole (DAPI) was used to label nuclear DNA. The samples were imaged by confocal microscopy (Carl Zeiss, LSM 880, Jena, Germany).

### 2.5. Alkaline Phosphatase Staining

The adherent cells in 6-well plate were washed three times with PBS and fixed by adding 4% paraformaldehyde for 10 min. The BCIP/NBT working solution (Beyotime, shanghai, China) was prepared according to the proportion specified by the manufacturer’s instructions, and then added for staining for 30 min. The slides were washed in PBS three times and were then observed under a microscope (Olympus, Tokyo, Japan).

### 2.6. Lentivirus Transduction

For knockdown of human PDLIM5 expression, Lentivirus vectors (NM_006457) repressing *PDLIM5* (GAGCAACTACAGTGTGTCACT) were constructed and generated by Genechem (Shanghai, China) (Appendix A). In preliminary experiments (Appendix A), we determined the optimal conditions for lentivirus infection, including inoculation, infection volume, duration of treatment, and number of cells with multiple infections. To infect target cells, the virus was diluted in fresh medium in accordance with the transfected cell density ratio before transduction, and the virus infection enhancer HistransG P was added. Before infection, the cells were digested and inoculated on the 6-well plate at a density of 7000/cm². The cells were cultured at 37 °C for 16–24 h until the degree of confluence was 20–30%. On the second day, according to cell MOI (multiplicity of infection) and the virus titer, the corresponding amount of virus and the corresponding infection enhancement solution were added. After 12 h, the medium was changed to the regular growth medium. At 72–96 h, the expression efficiency of green fluorescent protein (GFP) was observed by Olympus microscopy (Tokyo, Japan).

### 2.7. Cell Proliferation Assay (CCK8 Assay)

Following three washes of the adherent cells using PBS, a 10% CCK8 working solution (Cell CountingKit-8) was prepared with serum-free DMEM. The culture plate was incubated in the incubator for 1 h, and the absorbance of the sample at 450 nm was determined by an enzyme-labeling instrument (Thermo Fisher, Thermo Scientific Multiskan FC, Waltham, MA, USA).

### 2.8. Wound Healing

The cells were inoculated in a 6-well plate and divided into three groups: Con, shScr, shPDLIM5, the corresponding density of inoculated cells in each group was 5000/cm^2^, 5000/cm^2^, and 6000/cm^2^ respectively. The cells were uncovered but immersed in growth media to make scratches on the plates with 200 μL pipette tips. The old medium was discarded and the cell was washed with PBS three times to remove the detached cells. Next, serum-free medium was added for culturing. The motor ability of the cells was observed and photographed after 0 h, 12 h, 24 h, and 48 h.

### 2.9. Transwell Migration Experiments

Migration chambers were positioned in a 24-well plate and 100 μL of serum-free medium was added to the chambers. Next, 200 μL of medium containing the cell suspension (2.5 × 10^5^ cells/mL in serum-free medium) was placed in the upper chamber and 750 μL culture medium (with FBS) was added to the lower chambers. The plate was incubated at 37 °C for 12–16 h. Next, medium was removed from the chambers, the chambers were washed twice in PBS, and the cells were fixed in 4% formaldehyde for 30 min. The membranes were exposed to crystal violet staining solution (Solarbio Company, Beijing, China) for 30 min after washing in PBS. Non-migrated cells were scraped away with cotton swabs, and the migrated cells were observed under an Olympus microscope (Tokyo, Japan).

### 2.10. Statistical Analysis

All experiments were performed at least in triplicate. Data are presented as mean ± standard deviation (SD). *T*-test (GraphPad Prism 5.0 software, La Jolla, CA, USA) was used to determine the significant differences, and *p* < 0.05 was considered statistically significant.

## 3. Results

### 3.1. Cell Culture and Osteogenic Differentiation In Vitro

To verify the ability of osteogenic differentiation of HSFs, hASCs, which have been proved to be one of the representatives of mesenchymal stem cells with multiple differentiation functions and compared with hMSCs, hASCs have a wide range of sources, are easy to obtain, simple separation, and are not restricted by ethics, were induced to indirectly observe the osteogenic ability of fibroblasts. hASCs and HSFs completely adhered to the culture dishes 24 h after inoculation. Both cell types were similar in shape, and were characterized by long fusiform or oval shapes. The confluence of cells reached 80–90% after 36–48 h, and rapidly grew with a whirlpool-like appearance (Figure 1A). In the condition of osteogenic induction, Western blotting and qRT-PCR were used to detect the expression of osteogenic marker proteins of hASCs and HSFS as the induction time increased (Figure 1B,C). Furthermore, the osteogenic differentiation ability of HSFs and hASCs was confirmed by ALP staining, which showed that alkaline phosphatase staining increased with the increase of osteogenic induction days, and reached the peak on the 14th day (Figure 1D). Moreover, the results of alizarin red staining showed that on the 21st day of osteogenic induction, HSFs were weaker than hASCs in bone mineralization nodules (Appendix A), which may be related to the different differentiation potentials of the two kinds of cells.

### 3.2. Microfilaments and Related Proteins Were Involved in Osteogenic Differentiation of HSFs

To examine the effects of microfilaments on the osteogenic differentiation potential of HSFs, we detected the expression of β-actin-, α-actinin1-, and cytoskeleton-related protein PDLIM5 by RT-qPCR and Western blotting analyses in the osteogenic differentiation of HSFs. The results showed that the expression of RUNX2, ALP, OPN, β-actin, α-actinin1, and PDLIM5 were significantly upregulated in osteogenic medium (OS) (Figure 2A,B). It is suggested that the osteogenic differentiation of HSFs can increase the expression of actin cytoskeleton and its related proteins (such as PDLIM5) (Figure 2A,B), thus affecting the changes of cytoskeleton. Further, it has also been speculated that cytoskeleton and related protein PDLIM5 may play a role in the osteogenic differentiation of HSFs. Thus, the cells were further treated with microfilament inhibitor (CytoD) and stabilizer (JAS) and cultured in OS to observe the effects of microfilament on the osteogenic differentiation potential of HSFs. Western blotting was used to evaluate the expression of β-actin and osteogenesis-associated proteins. The results showed that the expression levels of RUNX2, ALP, and β-actin were significantly upregulated by OS. In the OS + CytoD group, the expression levels of RUNX2, ALP, and β-actin were significantly inhibited, and in the OS + JAS group, the inhibitory effects in the OS + CytoD treatments on the osteogenic marker protein RUNX2 and other proteins were offset (Figure 2C). ALP staining at 14 and 21 days after osteogenic induction showed that the OS + CytoD group decreased the staining intensity, while the OS + JAS group increased the staining intensity (Figure 2D). These results indicated that the disassembly of microfilaments inhibited the osteogenic ability of HSFs, while the polymerization of microfilament cytoskeleton molecules can promote the osteogenic ability of HSFs.

The expression of PDLIM5 decreased during the depolymerization of microfilament cytoskeleton (Figure 2C). It has been speculated that PDLIM5 may play a synergistic role with the microfilament cytoskeleton to regulate the osteogenic differentiation ability of HSFs. Therefore, we examined the cellular localization of PDLIM5. Immunofluorescence analysis showed the co-localization of PDLIM5 and F-actin in osteogenic differentiation of HSFs (Figure 2E). Furthermore, immunofluorescence also showed that PDLIM5 and α-actinin1 specifically co-localized to stress fibers (Figure 2F). We hypothesized that PDLIM5 may mediate the functional changes of cytoskeletal microfilaments by binding with α-actinin1, and thus affect the osteogenic differentiation of HSFs. Nonetheless, the relationship between PDLIM5 and α-actinin1 needs to be further verified. To test this hypothesis, we evaluated the effects of PDLIM5 on the osteogenic differentiation of microfilaments and observed that down-regulation of PDLIM5 occurred in HSFs.

### 3.3. PDLIM5 Knockdown Inhibited the Proliferation, Movement, and Migration Ability of HSFs

To investigate the function of PDLIM5 in HSFS, a lentiviral vector with deletion of the *PDLIM5* target gene was transfected and integrated into HSFs, which down-regulated PDLIM5 expression in human fibroblasts. Green fluorescent protein (GFP) expression was observed by fluorescence microscopy and we determined that the transfection efficiency of lentivirus transfection was about 90% (Figure 3A). The knockdown of PDLIM5 in HSFs was further assessed by RT-qPCR, Western blotting, and immunofluorescence analyses (Figure 3B–D). The CCK8 assay, wound-healing assay, and cell migration tests using the transfected HSFs showed that after knocking down of PDLIM5, the proliferation and mobility of fibroblasts were significantly inhibited (Figure 3E–G). These results indicated that PDLIM5 was involved in the growth and differentiation process of HSFs and may affect the biological behavior of cells as a key regulatory factor.

### 3.4. PDLIM5 Knockdown Attenuated the Osteogenic Differentiation of HSFs Mediated by Microfilament

To examine the effects of PDLIM5 on osteogenic differentiation of HSFs, the expression of osteogenesis-associated genes was assessed by RT-qPCR and Western blotting analyses after knocking down *PDLIM5*. The results showed that, compared with the negative control group (shScr), the expression levels of osteogenic marker protein and mRNA in the down-regulated PDLIM5 group (shPDLIM5) were significantly decreased, and protein and mRNA levels of β-actin and α-actinin1 also decreased after knock-down of PDLIM5 expression (Figure 4A,B). Consistent with these results, PDLIM5 knockdown weakened the alkaline phosphatase (ALP) staining during osteogenic differentiation of HSFs (Figure 4C). In addition, we also found that knocking down *PDLIM5* not only inhibited the expression of actin on osteogenesis, but also changed the morphology of microfilaments. Immunofluorescence analysis at 7 days post-osteogenic induction of HSFS showed that, compared with the shScr negative control group, the microfilaments in the shPDLIM5 group changed from their original neatly arranged filaments to short, wide, and thick filaments (Figure 4(Db)), while in the uninduced group, the morphology of microfilaments did not change before or after the down-regulation of PDLIM5 expression (Figure 4(Da)). These results suggest that PDLIM5 may exert a similar role as the depolymerizing agent of microfilaments, thereby inhibiting the expression of microfilaments and mediating changes in microfilaments stability so as to regulate the osteogenic differentiation of cells.

### 3.5. Stable Microfilaments Reversed the Inhibitory Effect of Knockdown PDLIM5 on Osteogenic Differentiation of HSFs

To determine the effects of PDLIM5 on the microfilament skeleton, the cells were treated with a microfilament stabilizer (JAS) and treated in OS medium for 7 days with or without *PDLIM5*-knockdown. Western blotting analysis showed that JAS reversed the inhibition of knocked-down PDLIM5 on ALP, RUNX2, β-actin, α-actin 1, and PDLIM5 levels (Figure 4E). ALP staining showed that the JAS reversed the inhibitory effect of PDLIM5 knockdown on HSFs osteogenesis (Figure 4F). Taken together, these results suggested that PDLIM5 was involved in mediating osteogenic differentiation of fibroblasts by influencing the expression of microfilament cytoskeleton proteins and morphological changes.

### 3.6. PDLIM5-Knockdown Inhibited the Nuclear Localization of YAP

In the process of HSFs osteogenesis induced by chemical stimulation, we found that YAP protein expression tended to increase (Figure 5A). Immunofluorescence analysis showed that the distribution of YAP protein in the GM group was reduced, and the nuclei appeared not clearly defined, while in the OS 7d treatment group, the nuclei containing YAP protein were increased, and the nuclei were clearly formed (Figure 5B). These results indicated that YAP could be induced to enter the nucleus in the osteogenic environment of HSFs. Therefore, to further understand the role of YAP in osteogenesis of HSFs, we examined the expression of YAP during the osteogenic differentiation of HSFs by knocking-down *PDLIM5* expression. Western blotting analysis showed that the nuclear YAP protein expression level in the shPDLIM5 group was significantly decreased compared with the negative control shScr group (Figure 5C). Immunofluorescence analysis showed that nuclear YAP expression was lower in the shPDLIM5 group, in which the nuclear shape and outline were not clearly defined, while in the blank control group and empty plasmid control (shScr) group the nuclear YAP expression was marked, and the nuclear shape was clearly demarcated (Figure 5D). The results indicated that knockdown of PDLIM5 expression reduced the nuclear localization of YAP in osteogenic induction of HSFs.

## 4. Discussion

Fibroblasts present the same morphology as human mesenchymal stem cells (hMSC) and are derived from many different tissues. Studies have shown that fibroblasts express the same markers as hMSC and have the ability to differentiate into different lineages [41,42]. Fibroblasts have attracted attention for their potential application in bone tissue engineering [10]. In this study, the role of PDLIM5 in osteogenic differentiation of HSFs was detected, and the potential mechanisms involving PDLIM5-F-actin interaction in osteogenic differentiation were identified, as was the effect of PDLIM5 on YAP activity by osteogenic signals.

A few studies are available on investigating the molecular mechanism of osteogenic differentiation of fibroblasts. Actin is the most abundant cytoskeleton protein and is considered to be a key factor in mediating intracellular and extracellular signal responses [43,44]. Studies have shown that microfilaments mediate the osteogenic differentiation of stem cells and regulate the transduction of the osteogenic signal pathway [45,46]. However, the mechanism involved in microfilaments regulating osteogenic differentiation of fibroblasts is not completely clear. In this study, microfilaments in the depolymerized state significantly inhibited the osteogenic ability of HSFs, while polymerization could promote the osteogenic ability of HSFs. This is consistent with the results of osteogenic differentiation of stem cells mediated by microfilaments [47]. Interestingly, during the depolymerization of the microfilament skeleton, the expression of PDLIM5 in osteogenic differentiation also decreased. Western blotting and qRT-PCR also showed that protein and mRNA levels of PDLIM5 were increased during the osteogenic differentiation of HSFs, suggesting that PDLIM5 may act as a regulatory factor affecting the osteogenic ability of cells.

Most studies have shown that PDLIM5 and its family members play key roles in the differentiation and functioning of various cell types through PDZ and LIM domains. For example, RIL is an actin-associated protein that stimulates actin binding by interacting with the actin cross-linking protein α-actinin1 to increase its affinity with filamentous actin [48]. Following binding to α-actinin, the PDZ-LIM protein CLP-36 was found to be localized in the actin stress fibers via its PDZ domain [49]. These results revealed the relationship between PDZ-LIM protein and microfilaments. As a member of the PDZ-LIM family, PDLIM5 was clearly shown to be an actin skeleton-related protein. It plays an indispensable role in the functional activity of cell biology [27,39]. In the present study, knocking down PDLIM5 significantly inhibited the proliferation, movement, and migration of fibroblasts, suggesting that PDLIM5 participates in the biological function of HSFs. To verify the role of PDLIM5 in osteogenic differentiation of HSFs, we examined the effects of *PDLIM5* knockdown on the expression of osteogenesis-related genes. The results showed that knockdown of PDLIM5 significantly inhibited the expression of β-actin and osteogenesis-related genes. Furthermore, *PDLIM5* knockdown also changed the morphology of microfilaments, suggesting that PDLIM5 may mediate the osteogenic differentiation of HSFs by affecting the activity of microfilaments. To further investigate the effects of PDLIM5 on microfilaments of the cytoskeleton, cells were treated with a microfilament stabilizer after lentivirus down-regulation of *PDLIM5* expression. The results showed that the microfilament stabilizer could reverse the inhibitory effect of missing *PDLIM5* expression on osteogenesis. Thus, these findings suggested that PDLIM5 may mediate the osteogenic differentiation of fibroblasts by affecting the expression of microfilament cytoskeleton protein and inducing morphological changes. In this study, we preliminarily discussed the role of PDLIM5 in chemical osteogenesis induction. The specific signaling pathway of PDLIM5-mediated microfilaments involved in osteogenic differentiation needs to be further evaluated in future research. In addition, as a mechanical sensitive element, how PDLIM5 plays a role in other mechanical stimuli is the direction we need to further explore. Therefore, our next step is to investigate how PDLIM5 participates in the mechanical signal transduction of osteogenic differentiation through cytoskeleton under mechanical stimulation by applying mechanical stimulation under Flexcell tensiometer. 

We also showed that knockdown of *PDLIM5* significantly reduced the expression of YAP in the nucleus and its nuclear localization. The YAP pathway has been shown to be involved in the process of cell osteogenesis [50,51]. YAP signaling is one of the more studied and complex pathways involved in osteogenic differentiation. It has been reported that F-actin polymerization enhances the expression of RhoA and transcription factor YAP/TAZ, which promotes the osteogenic differentiation of mesenchymal stem cells [52]. In addition, through the response to different ECM stiffness, Yap and TAZ move in and out of the nucleus under the control of MT1-MMP, where YAP/TAZ can be activated on the hard matrix to make the cells differentiate into osteoblasts [53,54]. Thus, we hypothesized that PDLIM5 may mediate the transcriptional activity of the key osteogenic transcription factor RUNX2 through the regulation of YAP access to the nucleus, thereby participating in the osteogenic differentiation of cells. As a protein closely related to the actin skeleton, PDLIM5 needs further research on YAP signaling to improve the influence of the signal axis of PDLIM5-F-Actin-YAP on the osteogenic differentiation of fibroblasts. 

In conclusion, the present study showed that human skin fibroblasts present an osteogenic differentiation potential as adipose stem cells, but the osteogenic differentiation of fibroblasts may be delayed and the bone-forming capacity is lower than hASCs. This potential can be reduced by the effect of PDLIM5 on the expression and morphology of microfilaments. These data provide more theoretical basis for the study of the seed source of fibroblasts in the field of bone tissue engineering, and propose novel ideas for the study of bone tissue engineering.

## Figures and Tables

**Figure 1 biomolecules-11-00759-f001:**
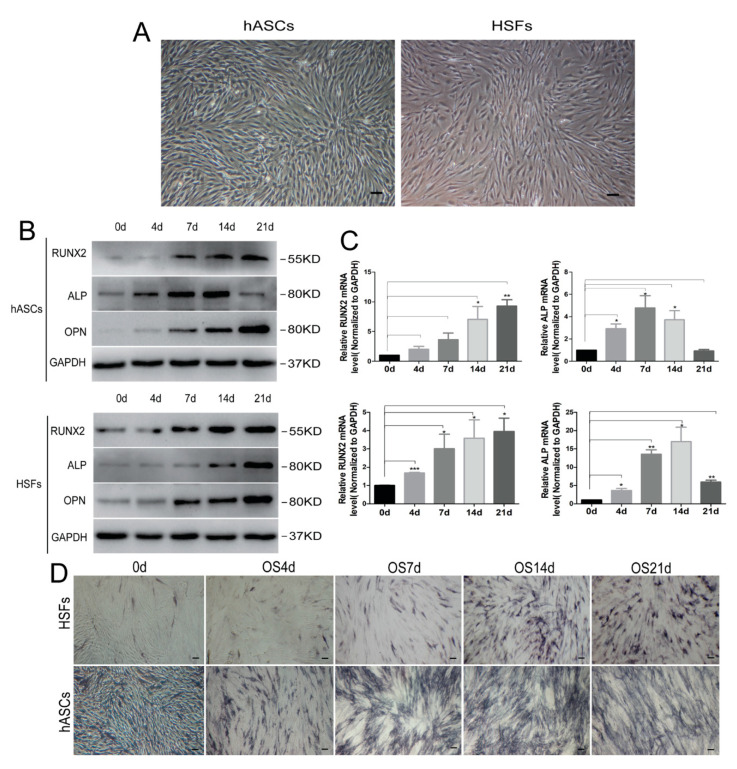
Morphology and osteogenic differentiation of hASCs and HSFs in vitro. (**A**) Morphology of hASCs and HSFs obtained following 48 h of culture. Scale bar = 100 µm. Western blotting (**B**) and RT-qPCR (**C**) detection of HSFs, and hASCs osteogenic differentiation marker protein and mRNA expression. (**D**) ALP staining analysis of hASCs and HSFs differentiation into osteogenesis. Scale bar = 100 µm. hASCs: human derived adipose stem cells; HSFs: human skin fibroblasts; OS: osteogenic medium; RUNX2: runt related transcription factor 2; ALP: alkaline phosphatase; OPN: osteopontin, * *p* < 0.05, ** *p* < 0.01, *** *p* < 0.001, n = 3.

**Figure 2 biomolecules-11-00759-f002:**
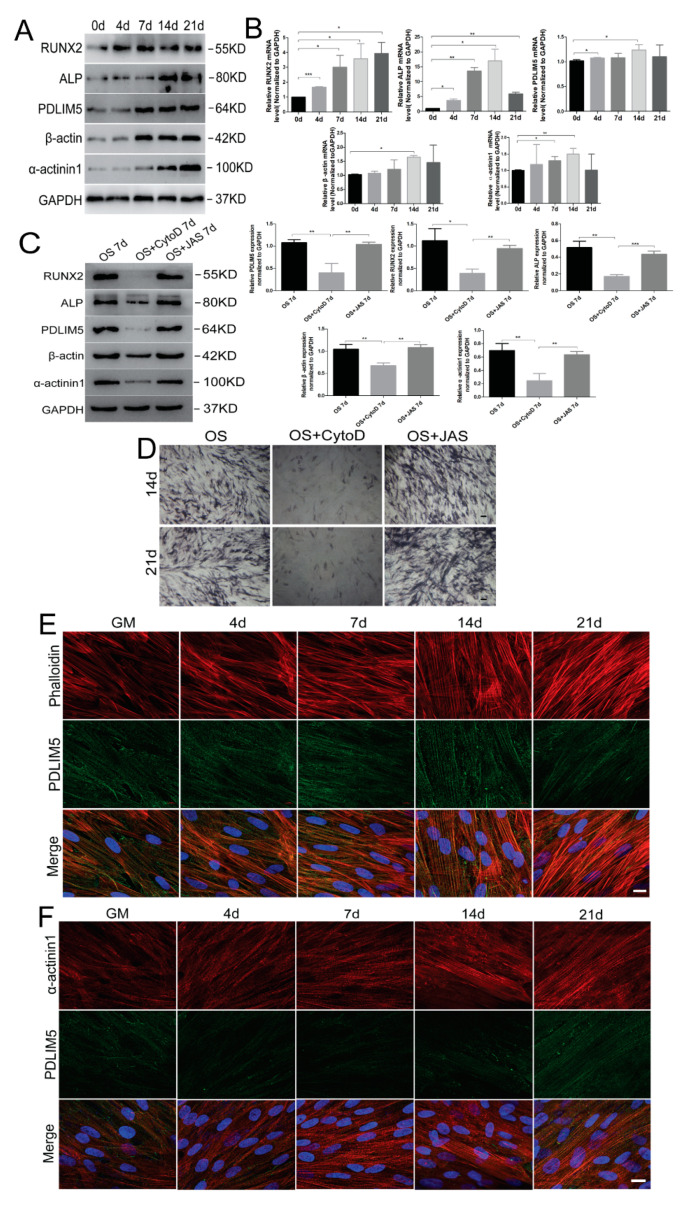
Microfilaments and related proteins are involved in osteogenic differentiation of HSFs. Detection of β-actin, α-actinin1, PDLIM5, and osteogenesis-associated gene expression by (**A**) Western blotting and (**B**) RT-qPCR analyses of HSFs treated with osteogenic differentiation medium (OS) on 0 d, 4 d, 7 d, 14 d, and 21 d. Effects of CytoD and JAS on the expression of HSFS osteogenic-associated protein and PDLIM5 by (**C**) Western blotting analyses. OS 7d, OS + CytoD, and OS + JAS 7d: HSFs treated with osteogenic differentiation medium, osteogenic differentiation medium containing 0.1 μg/mL of CytoD, and osteogenic differentiation medium containing 10 nM/mL of JAS for 7 days, respectively. Densitometric quantification of the Western blotting bands normalized to GAPDH. * *p* < 0.05, ** *p* < 0.01, *** *p* < 0.001, n = 3. Osteogenic differentiation of HSFs with CytoD or JAS in OS was determined by (**D**) ALP staining at 14 and 21 days. Scale bar = 100 µm. Immunofluorescence (**E**) detected the co-localization of PDLIM5 and F-actin, and (**F**) analysis detected that PDLIM5 co-localized specifically with α-actinin1 on key stress fibers during osteogenic differentiation. Scale bar = 10 µm. Phalloidin: F-actin immunofluorescent dye; CytoD: inhibitors of actin polymerization; JAS: actin polymerization agonist.

**Figure 3 biomolecules-11-00759-f003:**
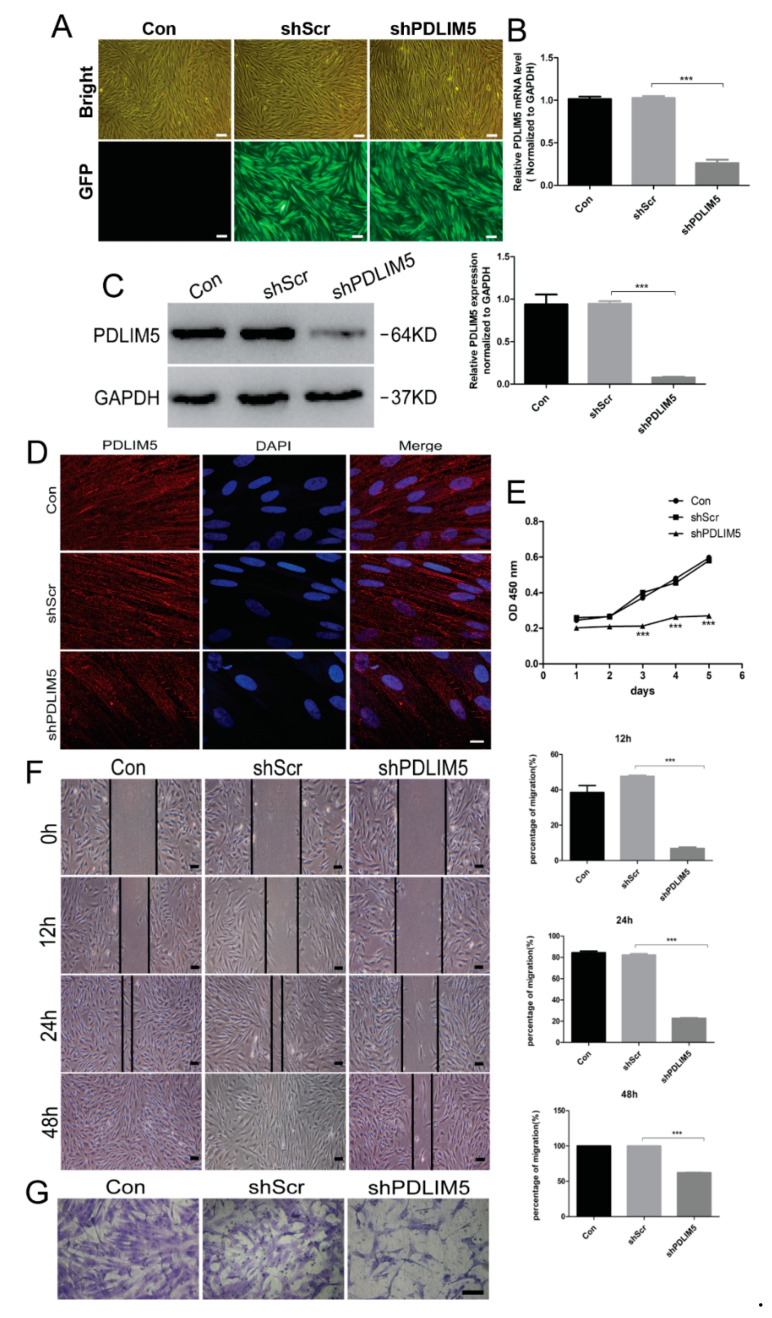
*PDLIM5* knockdown inhibits the proliferation, movement, and migration ability of HSFs. (**A**) HSFs fluorescence intensity 5 days after virus infection. Scale bar = 100 µm. Gene expression of *PDLIM5* by (**B**) RT-qPCR and (**C**) Western blotting analyses after lentivirus infection. *** *p* < 0.001, n = 3. The expression of knockdown PDLIM5 was detected by (**D**) immunofluorescence. Scale Bar = 10 µm. Knockdown of PDLIM5 inhibits HSFs proliferation (**E**), movement (**F**) and migration (**G**). *** *p* < 0.001, n = 3. Scale Bar = 100 µm. Con: blank control group: shScr: empty plasmid negative control group; shPDLIM5: experimental group of knock-down PDLIM5.

**Figure 4 biomolecules-11-00759-f004:**
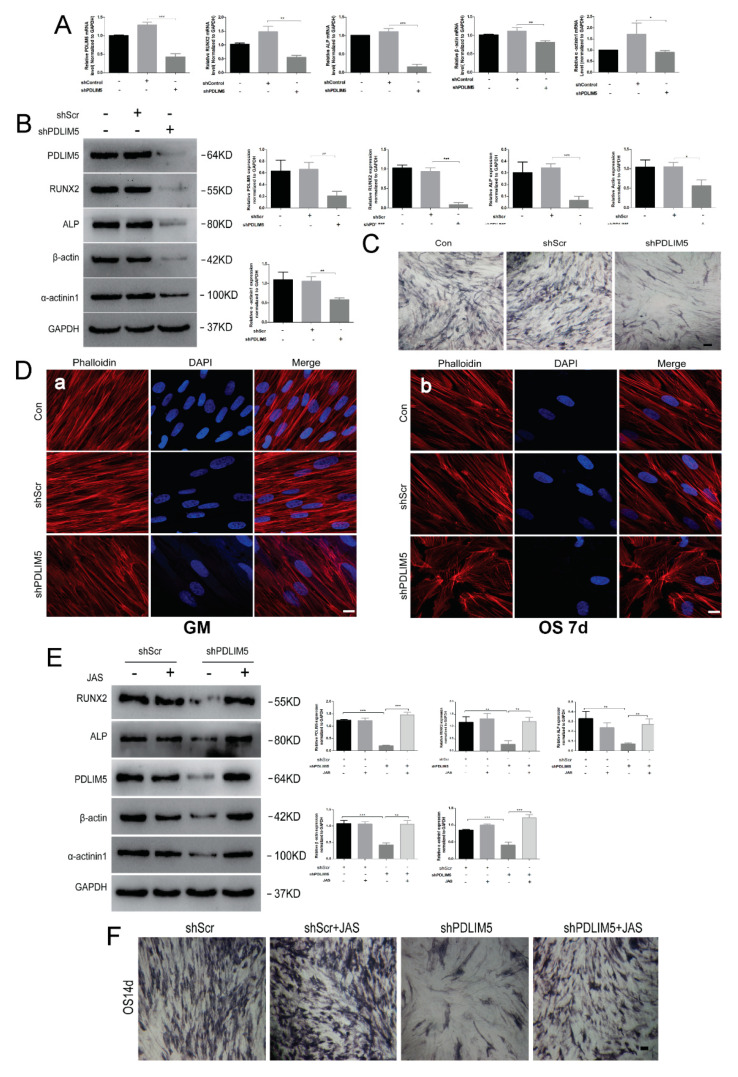
*PDLIM5* knockdown attenuates the osteogenic differentiation of HSF mediated by microfilaments. HSFs treated with osteogenic differentiation medium (OS) with or without knock-down of PDLIM5 at 7 days. β-actin and osteogenesis-associated gene expression by (**A**) RT-qPCR and (**B**) Western blotting analyses. Densitometric quantification of the Western blotting bands normalized to GAPDH. * *p* < 0.05, ** *p* < 0.01, ****p* < 0.001, n = 3. ALP staining (**C**) at 14 days. Scale bar = 100 µm. The effect of knocking down PDLIM5 on the morphology of microfilament skeleton was detected by immunofluorescence (**D**), a: knockdown *PDLIM5* in GM (undifferentiated) group; b: knockdown *PDLIM5* in osteogenic medium group for 7 days, Scale bar = 10 µm. (**E**) The effect of JAS on the osteogenic differentiation of HSF after knocking down *PDLIM5*. Densitometric quantification of the Western blot normalized to GAPDH. * *p* < 0.05, ** *p* < 0.01, *** *p* < 0.001, n = 3. ALP staining (**F**) at 14 days of HSFs osteogenic medium with JAS after knocking down *PDLIM5* expression. Scale bar = 100 µm.

**Figure 5 biomolecules-11-00759-f005:**
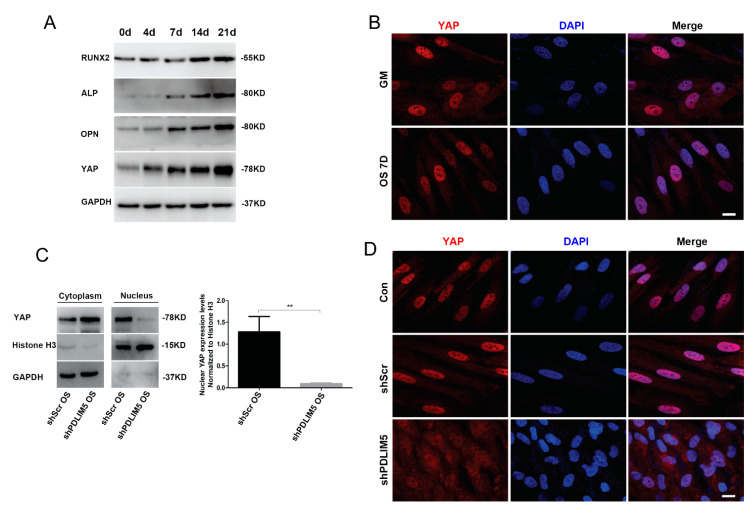
*PDLIM5* knockdown inhibited nuclear localization of YAP. YAP gene expression by (**A**) Western blotting analyses in osteogenic medium (OS) of HSFs at 0 d, 4 d, 7 d, 14 d, and 21 d. (**B**) The expression of nuclear YAP in HSFs with GM or OS for 7 days by immunofluorescence. The expression of nuclear YAP in HSFs treated with osteogenic differentiation medium with or without knocking down PDLIM5 at 7 days by (**C**) Western blotting analyses. Densitometric quantification of the Western blotting bands normalized to Histone 3. ** *p* < 0.01, n = 3. PDLIM5-knock-down reduced the nuclear localization of YAP and was detected by (**D**) immunofluorescence. Scale bar = 10 µm.

**Table 1 biomolecules-11-00759-t001:** Primers used in the qPCR.

Gene	Forward Primer (5′→3′)	Reverse Primer (3′→5′)
RUNX2	GCCTTCAAGGTGGTAGCCC	CGTTACCCGCCATGACAGTA
ALP	CTATCCTGGCTCCGTGCTC	CGCCAGTACTTGGGGTCTTT
PDLIM5	TTAGTGGCACTGGGGAAATC	GATCTTCCTTTGGCATCGAC
β-actin	CTTCGCGGGCGACGAT	CCACATAGGAATCCTTCTGACC
α-actinin1	ATTGGCAACGACCCCCAGAA	ATGTTGTAACCCATGGAGATCAGG
GAPDH	TCGGAGTCAACGGATTTGGT	TTCCCGTTCTCAGCCTTGAC

## Data Availability

All the supporting data can be downloaded.

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
