# Peer review of "Mechanical Sensing Element PDLIM5 Promotes Osteogenesis of Human Fibroblasts by Affecting the Activity of Microfilaments"

_biomolecules, 2021, doi:10.3390/biom11050759_

Round 1

Reviewer 1 Report

The authors of the study investigated the role of the mechanosensitive element, PDLIM5 in skin fibroblasts undergoing osteogenic differentiation. The results of the study demonstrate that inhibition of PDLIM5 via shRNA significantly inhibited osteogenic differentiation of skin fibroblasts. This was shown via a significant reduction in ALP and osteopontin, whilst knockdown of these microfilaments also resulted in inhibiting migration, proliferation and nuclear localization of YAP. The control for the study was adipose derived MSCs.

The results of investigation are interesting and provide further information on osteogenic pathway induction for skin fibroblasts. The authors of the study should address the following points prior to publication of the study.

  1. Are the skin fibroblasts and adipose derived stem cells different apart from their origin ? The authors should provide CD marker characterisation or gene expression/western blots for stem cell markers for both skin fibroblasts and adipose derived stem cells to see if there are similarities or differences between the cell types.
  2. Though the focus is primarily on osteogenesis, have the authors investigated other musculoskeletal lineages to understand whether inhibition PDLIM5 leads to induction of either adipogenic or chondrogenic lineages ? Gene expression or western blot to see if PDLIM5 shRNA skin fibroblasts express markers for these lineages should be provided. Many studies have shown that control of cytoskeleton influences MSC differentiation, particularly chondrogenesis.
  3. The authors in their introduction state about the RAS-ERK signalling pathway. The authors should present data with respect to genes within this pathway and ERK western blot to demonstrate the other downstream effect of shRNA inhibition of PDLIM5 apart from inhibited YAP nuclear localisation (e.g. expression of GTPases, RhoA, ROCK etc.)
  4. What do the authors hypothesize to be the pathway to be influenced by PDLIM5 inhibition in osteogenesis ? Is this shear, compression, tensile etc. ? Authors should hypothesize the mechanical stimuli influenced by inhibiting PDLIM5 in their discussion.

Author Response

The authors of the study investigated the role of the mechanosensitive element, PDLIM5 in skin fibroblasts undergoing osteogenic differentiation. The results of the study demonstrate that inhibition of PDLIM5 via shRNA significantly inhibited osteogenic differentiation of skin fibroblasts. This was shown via a significant reduction in ALP and osteopontin, whilst knockdown of these microfilaments also resulted in inhibiting migration, proliferation and nuclear localization of YAP. The control for the study was adipose derived MSCs.

The results of investigation are interesting and provide further information on osteogenic pathway induction for skin fibroblasts. The authors of the study should address the following points prior to publication of the study.

Response: Thank you for your comment.

Are the skin fibroblasts and adipose derived stem cells different apart from their origin? The authors should provide CD marker characterisation or gene expression/western blots for stem cell markers for both skin fibroblasts and adipose derived stem cells to see if there are similarities or differences between the cell types.

Response: Thank you for your comment. Skin fibroblasts are derived from human dermis, while adipose stem cells are derived from subcutaneous adipose tissue. Since a large number of literatures have reported the verification of the expression of two kinds of cell-related surface markers, they have not been repeated in the study. Two kinds of cells are similar in morphology, and the surface markers are positive for CD29, CD44, CD73, CD90, CD105, CD166, and negative for CD14, CD31, CD34, CD45, CD133, etc. The cells can differentiate into osteogenesis, chondrogenesis and adipogenesis. This paper focuses on the study of osteogenic differentiation, which provides that both kinds of cells can express osteogenic markers under the condition of osteogenic induction. Although two kinds of cells are similar in multi-directional differentiation, they are different in angiogenesis and other functions. In this study, it is found that the osteogenic ability of fibroblasts is slightly longer than that of adipose stem cells, so it is difficult to distinguish between two kinds of cells in terms of markers and functions. The purpose of using adipose stem cells in this paper is to indirectly verify whether the fibroblasts have similar osteogenic differentiation ability, and to lay a foundation for the follow-up study on the mechanism of multiple differentiation of fibroblasts. The verification of the lack of relevant surface markers is our deficiency, and we will further verify this function in the follow-up research. The following is the support of the relevant literature on the study of two kinds of cell surface markers:

(1) Mitchell J B, Mcintosh K, Zvonic S, et al. Immunophenotype of human adipose-derived cells: temporal changes in stromal associated and stem cell-associated markers. Stem Cells, 2006,24(2):376~385.

(2) Gronthos S, Franklin D M, Leddy H A, et al. Surface protein characterization of human adipose tissue-derived stromal cells. J Cell Physiol,2001,189(1):54~63.

(3) Lorenz K, Sicker M, Schmelzer E, et al. Multilineage differentiation potential of human dermal skin-derived fibroblasts. Exp Dermatol,2008,17(11):925~932.

(4) Soundararajan M, Kannan S. Fibroblasts and mesenchymal stem cells: Two sides of the same coin? J Cell Physiol,2018, 233(12):9099~9109.

(5) Blasi A, Martino C, Balducci L, et al. Dermal fibroblasts display similar phenotypic and differentiation capacity to fat-derived mesenchymal stem cells, but differ in anti-inflammatory and angiogenic potentiall. Vasc Cell,2011,3(1):5.

(6) Brohem C A, de Carvalho C M, Radoski C L, et al. Comparison between fibroblasts and mesenchymal stem cells derived from dermal and adipose tissue. Int J Cosmet Sci, 2013,35(5): 448 ~457.

Though the focus is primarily on osteogenesis, have the authors investigated other musculoskeletal lineages to understand whether inhibition PDLIM5 leads to induction of either adipogenic or chondrogenic lineages ? Gene expression or western blot to see if PDLIM5 shRNA skin fibroblasts express markers for these lineages should be provided. Many studies have shown that control of cytoskeleton influences MSC differentiation, particularly chondrogenesis.

Response: Thank you for your comment. You are very far-sighted and can see the direction of our follow-up research. At present, this study is only in the direction of osteogenesis to verify the effect of PDLIM5 on the osteogenic ability of cells through the cytoskeleton, we have not confirmed the role of PDLIM5 in other differentiation lineages, which is our follow-up work.

The authors in their introduction state about the RAS-ERK signalling pathway. The authors should present data with respect to genes within this pathway and ERK western blot to demonstrate the other downstream effect of shRNA inhibition of PDLIM5 apart from inhibited YAP nuclear localisation (e.g. expression of GTPases, RhoA, ROCK etc.)

Response: Thank you for your comment. The RAS-ERK signal pathway involved in the background introduction is mainly about the pathway related to the effect of PDLIM5 on cell proliferation, while the main purpose of this study is to explore the molecular mechanism that PDLIM5 regulates the osteogenic ability of cells through microfilament cytoskeleton. At the same time, in the course of the study, it is found that inhibition of PDLIM5 can reduce the nuclear localization of YAP. The lack of related downstream effect pathway is the defect of this paper. Other specific downstream effects are also taken into account in the follow-up study.

What do the authors hypothesize to be the pathway to be influenced by PDLIM5 inhibition in osteogenesis ? Is this shear, compression, tensile etc.? Authors should hypothesize the mechanical stimuli influenced by inhibiting PDLIM5 in their discussion.

Response: Thank you for your comment. In this study, we mainly study the regulatory role of PDLIM5 in chemically induced osteogenesis, and PDLIM5 as a mechanical sensitive element. Our next step is to apply mechanical stimulation under Flexcell extensometer to verify the role of PDLIM5 in mechanically stimulated osteogenesis. The lack of this hypothesis in the discussion is our mistake, which has been explained in the discussion part of the article in line 437-444 and marked it in red according to your suggestion.

Reviewer 2 Report

This is an interesting work highlighting the importance of PDLIM5 in osteogenic differentiation of human fibroblasts. The authors report that PDLIM5 may have a role in microfilament polymerisation and, thus, promote osteogenic differentiation of fibroblasts. They used a variety of experimental approached to elucidate this concept. However, after careful considerations, many points need to be addressed. In addition, language has to be edited.

  • Lines 24, 25, 29: Use osteogenic differentiation instead of osteogenesis.
  • Line 31: Use formation instead of expression.
  • Lines 38-39: Needs rephrasing, repetition.
  • Lines 41-43: This sentence means that human fibroblasts can go back, become stem cells and redifferentiate. Please rephrase.
  • Line 45: “play the same role” is too vague. Please rephrase.
  • Line 53: “to search for the reactive protein of protein kinase C” needs rephrasing.
  • Lines 60-61: This is unclear. Which integrin? which pathway?
  • Lines 63-65: Myogenesis and myotube formation follow specific and well-defined steps. In which PDLIM5 is involved?
  • Line 74: Is cytoskeleton an organelle?
  • Lines 76-77: These are the basic structures of cytoskeleton, not cytoplasm.
  • Lines 78-83: This sentence is too long.
  • Line 82: This is not clear. For what is Runx2 a target gene?
  • Lines 83-86: This is written earlier, repetition.
  • Line 106: Citation is needed. Briefly describe the collagenase digestion method.
  • Line 111: What was the number of cells?
  • Line 113: OSM is comprised of 50mg/L ascorbic acid. Why the authors used 37.5mg/L?
  • Why two different PDLIM5 Abs were used in Western blotting?
  • Immunofluorescence: Were the cells used for IF still in petri dishes?
  • ALP staining: These slides that were used are not referred earlier. Not clear. Were cells grown on slides?
  • Line 196: What kind of plate? 96-well?
  • Lentivirus transfection: What was the MOI? Was cytotoxicity assessed? Preliminary experiments should be presented as supplementary material.
  • Line 201: GFP expression is not specified. The structure of the vector should be described in detail.
  • Line 210: How many cells? Which groups?
  • Line 212: “cell surface” needs rephrasing.
  • Lines 234-235: The use of hASCs has to be justified in detail. Why did the authors select this type instead of hMSCs?
  • Osteogenic differentiation section and Fig1: After 21d in OSM, osteoblasts have started to form bone nodules. The bone formation should be also evaluated and could be used to prove the osteoblastic differentiation. Also, CFU-OB could be applied. Runx2 expression is found increased in the first stages of osteoblastic differentiation and starts to decrease when mature osteoblasts have been formed. The appropriate markers for late stages are osteocalcin (or Bglap expression) and Osx. Please comment.
  • Throughout the entire text: Gene and protein names (italics and capitals) should be distinguished.
  • Lines 255-256: “It has been suggested…”. Citation needed.
  • Line 324: Please define osteogenic ability.
  • Lines 433-437: This is too vague. Please rephrase.
  • Lines 441-444: Needs rephrasing.
  • Lines 445-446: This conclusion cannot been drawn by this set of experiments. No direct comparison was applied and Fig1D as well as Alp expression show the opposite.

Author Response

This is an interesting work highlighting the importance of PDLIM5 in osteogenic differentiation of human fibroblasts. The authors report that PDLIM5 may have a role in microfilament polymerisation and, thus, promote osteogenic differentiation of fibroblasts. They used a variety of experimental approached to elucidate this concept. However, after careful considerations, many points need to be addressed. In addition, language has to be edited.

Response: Thank you for your comment.

Lines 24, 25, 29: Use osteogenic differentiation instead of osteogenesis.

Response: Thank you for your suggestion. I have replaced osteogenesis with osteogenic differentiation at line 24, 25 and 30, and marked them in red.

Line 31: Use formation instead of expression.

Response: Thank you for your suggestion. I have modified according to your suggestion in line 31 and marked it in red.

Lines 38-39: Needs rephrasing, repetition.

Response: Thanks. I have deleted and modified according to your suggestion in line 37-38 and marked it in red.

Lines 41-43: This sentence means that human fibroblasts can go back, become stem cells and redifferentiate. Please rephrase.

Response: Thanks. I have deleted and modified according to your suggestion in line 40-42 and marked it in red.

Line 45: “play the same role” is too vague. Please rephrase.

Response: Thanks. I have rephrased it according to your suggestion in line 43 and marked it in red.

Line 53: “to search for the reactive protein of protein kinase C” needs rephrasing.

Response: Thank you for your comment. I have deleted and modified according to your suggestion in line 50-51 and marked it in red.

Lines 60-61: This is unclear. Which integrin? which pathway?

Response: Thanks. I have described the integrin and related signal pathways in the article based on your suggestions and insert the corresponding literature, and marked it in red in line 58-60.

Lines 63-65: Myogenesis and myotube formation follow specific and well-defined steps. In which PDLIM5 is involved?

Response: Thanks. This is my negligence and did not express it clearly. I have checked and corrected the sentence and marked it in red in line 60-63.

Line 74: Is cytoskeleton an organelle?

Response: Thank you for your comment. It's my mistake that didn't make it clear. I have replaced the organelles with cellular components and marked it in red in line 72.

Lines 76-77: These are the basic structures of cytoskeleton, not cytoplasm.

Response: Thanks. It's my mistake and I have corrected it in line 74 and marked it in red.

Lines 78-83: This sentence is too long.

Response: Thank you for your comment. I have made changes in the article and marked it in red in line 76-82.

Line 82: This is not clear. For what is Runx2 a target gene?

Response: Thanks. It's my mistake that didn't make it clear I have checked it in line 80 and marked it in red.

Lines 83-86: This is written earlier, repetition.

Response: Thanks. I have modified and deleted the repetitive sentences in line 82-83 according to your suggestion and marked it in red.

Line 106: Citation is needed. Briefly describe the collagenase digestion method.

Response: Thanks. I have quoted the corresponding literature and briefly described the collagenase digestion method in line 103-108 according to your suggestion and marked it in red.

Line 111: What was the number of cells?

Response: Thank you for your comment. I have added the cell density in the article in line 114 and marked it in red.

Line 113: OSM is comprised of 50mg/L ascorbic acid. Why the authors used 37.5mg/L?

Response: Thank you very much. It has been checked repeatedly that this concentration has been found out by our research group for many years to be suitable for osteogenic differentiation of our cells, and relevant articles have been successfully published with this concentration.

Why two different PDLIM5 Abs were used in Western blotting?

Response: Thanks. Since the purchased Abcam antibody is not suitable for immunofluorescence and is mainly used for western blotting, another Abnova antibody suitable for immunofluorescence was purchased, and it was verified that the western blotting results of the two antibodies were consistent.

Immunofluorescence: Were the cells used for IF still in petri dishes?

Response: It was my carelessness that I didn't give a clear description of the details. The cells were inoculated on the climbing plate suitable for 24-well plate. I have explained it in line 172 and marked in red.

ALP staining: These slides that were used are not referred earlier. Not clear. Were cells grown on slides?

Response: It was my carelessness that I didn't give a clear description of the details. The cells were planted on a 6-well plate and stained after adhesion and chemical osteogenic induction, and the cells were still on the culture plate at the time of staining. I have revised it and marked it red in line 187-188.

Line 196: What kind of plate? 96-well?

Response: Thanks. It was my carelessness that I didn't give a clear description of the details. The cells were planted on a 6-well plate. I have revised it and marked it red in line 200.

Lentivirus transfection: What was the MOI? Was cytotoxicity assessed? Preliminary experiments should be presented as supplementary material.

Response: Thanks. MOI (multiplicity of infection) refers to the ability of the virus to infect cells. Preliminary experiments will be presented as supplementary material 3.

Line 201: GFP expression is not specified. The structure of the vector should be described in detail.

Response: According to your suggestion, the structure of PDILM5 virus vector will be presented as supplementary material 4 and the target sequence has been described in line 193-195 and marked it in red.

Line 210: How many cells? Which groups?

Response: It was my carelessness that I didn't give a clear description of the details. I have revised it and marked it red in line 214-216.

Line 212: “cell surface” needs rephrasing.

Response: Thank you for your comment. I have deleted and modified according to your suggestion in line 218 and marked it in red.

Lines 234-235: The use of hASCs has to be justified in detail. Why did the authors select this type instead of hMSCs?

Response: Thanks. hASCs have been proved to be one of the representatives of mesenchymal stem cells with multiple differentiation functions. Compared with hMSCs, hASCs have a wide range of sources, easy to obtain, simple separation, and are not restricted by ethics. At present, our research group is more familiar with the differentiation of hASCs, thus we choose hASCs as the early control in our study. I have explained it in line 239-242 according to your suggestion.

Osteogenic differentiation section and Fig1: After 21d in OSM, osteoblasts have started to form bone nodules. The bone formation should be also evaluated and could be used to prove the osteoblastic differentiation. Also, CFU-OB could be applied. Runx2 expression is found increased in the first stages of osteoblastic differentiation and starts to decrease when mature osteoblasts have been formed. The appropriate markers for late stages are osteocalcin (or Bglap expression) and Osx. Please comment.

Throughout the entire text: Gene and protein names (italics and capitals) should be distinguished.

Response: Osteopontin (OPN) is a rich protein in bone matrix, so OPN was selected as one of the marker proteins to verify osteogenic ability. As a key transcription factor in osteogenic differentiation, RUNX2 plays an important role in regulating the expression of genes responsible for osteogenic specific matrix proteins (including ALP, COLI, BSP, OCN, OPN), and ultimately stimulates bone nodular mineralization. In our study, on the 21st day of osteogenic differentiation, Runx2 continued to increase, which may be related to the ability of Runx2 to regulate specific matrix proteins in the middle and late stage. According to your suggestion, we will provide the results of alizarin red staining to observe calcium deposition after 21 days of osteogenic induction to further verify the osteogenic ability in late stages. Alizarin red staining will be presented as supplementary material 5 and marked it in red in line 252-253. Gene names (italics) have been distinguished in the entire text, and marked it in italic red in full text.

Lines 255-256: “It has been suggested…”. Citation needed.

Response: This is a mistake in my tense expression. This is the part that describes the findings of this study, and I have modified the sentence in line 266-267.

Line 324: Please define osteogenic ability.

Response: Thank you for your comment. The result of this part is that I have described it incorrectly and have reorganized and modified the statements according to your suggestion in line 334-336.

Lines 433-437: This is too vague. Please rephrase.

Response: Thanks. I have deleted and modified the sentence according to your suggestion and marked it in red in line 449-455.

Lines 441-444: Needs rephrasing.

Response: I have deleted and rephrased the sentence according to your suggestion and marked it in red in line 458-460.

Lines 445-446: This conclusion cannot been drawn by this set of experiments. No direct comparison was applied and Fig1D as well as Alp expression show the opposite.

Response: Thanks. The sentence has been modified and explained in line 461-463 and marked it in red according to your suggestion.

Round 2

Reviewer 2 Report

The authors addressed most of the points. However, Fig1 shows inconsistent results for ALP in both hASCs and HSFs. At 21d, HASCs express low levels of ALP gene, in comparison with 14d, which is in agreement with protein levels as determined by Western blotting but this is not reflected in the ALP staining. On the other hand, at 21d HSFs express low levels of ALP gene as compared to 14d which opposes the Western blot detection as well as the ALP staining. Furthermore, in Fig S5, it is shown that HSFs display a very low bone-forming capacity at 21d as compared to hASCs. Therefore, in line 263, the bone nodules are not "obvious" and, in lines 478-480, "similar" must be deleted and the "differentiation may be delayed" while "bone-forming capacity is considerably lower than hASCs". This sentence should be modified, accordingly.

Author Response

Point 1: The authors addressed most of the points. However, Fig1 shows inconsistent results for ALP in both hASCs and HSFs. At 21d, HASCs express low levels of ALP gene, in comparison with 14d, which is in agreement with protein levels as determined by Western blotting but this is not reflected in the ALP staining. On the other hand, at 21d HSFs express low levels of ALP gene as compared to 14d which opposes the Western blot detection as well as the ALP staining. Furthermore, in Fig S5, it is shown that HSFs display a very low bone-forming capacity at 21d as compared to hASCs. Therefore, in line 263, the bone nodules are not "obvious" and, in lines 478-480, "similar" must be deleted and the "differentiation may be delayed" while "bone-forming capacity is considerably lower than hASCs". This sentence should be modified, accordingly.

Response 1: Thank you for your comments. For the results of ALP staining in hASCs, there is no significant difference between 14 days and 21 days, but in fact, you can see that the ALP staining results of 14 days are denser than those of 21 days. It is also consistent with the results of protein and gene. As a marker of early and middle osteogenesis, the expression of ALP began to decrease after 14 days. However, the protein expression and gene level of ALP in HSFs was inconsistent at 21 days, which may be that the expression time of gene of ALP in fibroblasts was a little earlier than that of protein. And the expression of ALP gene began to decrease after 21 days. It is speculated that ALP protein will also show a downward trend, which may also be the reason why the osteogenic expression time of fibroblasts is a little longer than that of adipose-derived stem cells. With regard to the results of alizarin red staining, I did not describe clearly enough. I have re-explained it in the results section in line 262-265. The inappropriate sentences in the discussion section have been modified in line 479-481 according to your suggestion, and marked it in red.